# Effects of systemic inflammation on the network oscillation in the anterior cingulate cortex and cognitive behavior

**Ayumi Hirao[1], Yasushi Hojo[1] \*, Gen Murakami[2], Rina Ito[1], Miki Hashizume[1], Takayuki Murakoshi[1], Naonori Uozumi[1] \***

**1** Department of Biochemistry, Faculty of Medicine, Saitama Medical University, Moroyama, Iruma, Saitama, Japan, **2** Department of Liberal Arts, Faculty of Medicine, Saitama Medical University, Moroyama, Iruma, Saitama, Japan

\* nuozumi@saitama-med.ac.jp (NU); yhojo@saitama-med.ac.jp (YH)

**Data Availability Statement:** All relevant data are within the manuscript and its Supporting Information files.

**Funding:** The author(s) received no specific funding for this work.

## Abstract

Network oscillation in the anterior cingulate cortex (ACC) plays a key role in attention, novelty detection and anxiety; however, its involvement in cognitive impairment caused by acute systemic inflammation is unclear. To investigate the acute effects of systemic inflammation on ACC network oscillation and cognitive function, we analyzed cytokine level and cognitive performance as well as network oscillation in the mouse ACC Cg1 region, within 4 hours after lipopolysaccharide (LPS, 30 µg/kg) administration. While the interleukin-6 concentration in the serum was evidently higher in LPS-treated mice, the increases in the cerebral cortex interleukin-6 did not reach statistical significance. The power of kainic acid (KA)-induced network oscillation in the ACC Cg1 region slice preparation increased in LPS-treated mice. Notably, histamine, which was added *in vitro*, increased the oscillation power in the brain slices from LPS-untreated mice; for the LPS-treated mice, however, the effect of histamine was suppressive. In the open field test, frequency of entries into the center area showed a negative correlation with the power of network oscillation (0.3 µM of KA, theta band (3–8 Hz); 3.0 µM of KA, high-gamma band (50–80 Hz)). These results suggest that LPS-induced systemic inflammation results in increased network oscillation and a drastic change in histamine sensitivity in the ACC, accompanied by the robust production of systemic pro-inflammatory cytokines in the periphery, and that these alterations in the network oscillation and animal behavior as an acute phase reaction relate with each other. We suggest that our experimental setting has a distinct advantage in obtaining mechanistic insights into inflammatory cognitive impairment through comprehensive analyses of hormonal molecules and neuronal functions.

## Introduction

The anterior cingulate cortex (ACC), a component of the limbic system, is involved in the integration of cognitive processes, including attention [1], memory consolidation [2], novelty detection [3], arousal [4], empathy [5] and anxiety [6].

**Competing interests:** The authors have declared that no competing interests exist.

Network oscillations, rhythmic and synchronized activities generated by neuronal populations in neural circuits, underlie various coordinated functions [7, 8]. Theta oscillatory power in the mouse hippocampus has been reported to reflect learning and memory performance [9, 10]. Gamma oscillatory power in the frontal lobe has been shown to be decreased in schizophrenic patients [11, 12]. These findings suggest that ACC network oscillations play an important role in integrated functions such as attention and cognition and that oscillatory power serves as an indicator of cognitive status [13].

Lipopolysaccharide (LPS) acutely induces sickness behavior, mediated through production of pro-inflammatory cytokines, peaking at 3–4 hours [14, 15]. During the recovery from the acute sickness, higher cognitive performances, e.g. attention, and recognition of novel objects, are observed impaired at 24 hours or later in rats [16] and in mice [17]. However, it is not well understood whether LPS has an acute effect on higher cognitive functions, and how network oscillation and cognitive functions in the ACC could relate with each other within 3–4 hours. An increase in gamma oscillation power in the murine primary somatosensory cortex was reported using Complete Freund's Adjuvant-induced inflammation model [18].

Histamine is a neuromodulator that contributes to arousal in the CNS [19]. Histaminergic projections from the tuberomammillary nucleus cover various regions of the CNS and form the histaminergic system [19]. Four histamine receptor subtypes, H1, H2, H3, and H4, have been investigated, and the distribution of all subtypes has been confirmed in the CNS [20, 21]. H1 and H2 receptors are expressed postsynaptically in most brain regions [22, 23]. H3 receptors are presynaptic autoreceptors that regulate histamine release from nerve terminals [24] and are enriched in the cerebral cortex, striatum, cerebellum, and hippocampus [23]. H4 receptors are expressed in layer IV of the cerebral and entorhinal cortices [21]. Their gross distribution has been reported, however, their detailed distribution in the ACC remains unclear. Certain H1 antihistamine drugs with profound antiallergic effects are known to induce drowsiness through their action on H1 receptors in the CNS, indicating the contribution of the histaminergic system to arousal levels and cognitive performance. Increased histamine concentrations in the CNS have been reported in LPS-treated mice [25]. These findings raise the possibility that LPS affects cognitive function through histaminergic neurons of the ACC. We have been investigating network oscillations *in vitro* using rodent brain slices with a focus on the Cg1 region of ACC and find this region suitable for evaluating the pharmacological effects of histamine on network oscillations [26, 27].

The purposes of the present study were as follows:1) to clarify how acute systemic inflammation affects network oscillations in the ACC and 2) to demonstrate how changes in network oscillations are correlated with animal behavior and local cytokine production. Our acute inflammation model utilizes an intraperitoneal injection of LPS in mice. We analyzed the effects of acute LPS administration on 1) kainic acid (KA)-induced network oscillation from the Cg1 region of the ACC, either in the presence or absence of histamine; 2) anxiety by open field [6]; 3) novelty detection by novel object recognition test [2]; and 4) Interleukin (IL)-6 concentration in the serum and cerebral cortex as a surrogate marker for inflammation.

## Materials and methods

### Mice

All animal experiments were conducted in accordance with protocols approved by the Animal Experiment Committee of Saitama Medical University (Approval ID:3238).

Thirty-day-old male C57BL/6J mice were purchased from Tokyo Laboratory Animals Science (Tokyo, Japan) and acclimatized to the environment of the animal facility for one week before the experiment. The mice were individually housed in a temperature-controlled room

maintained on a 12 h light/dark cycle, with food and water available *ad libitum*. The mice were aged 35–42 days. Brain slices were prepared from the mice on the same day as the behavioral experiments.

### Novel object recognition test

The novel object recognition test (NOR) was conducted following the protocol by Jung *et. al* [28] and by Ennaceur *et. al* [29] with slight modifications. The day before the experiment, the mice were acclimated to the laboratory environment and allowed to explore an empty arena (length × width × height, 40 × 40 × 30 cm) for 10 min as the first habituation before the NOR test (Fig 1).

On the next day, 0.9% saline (n = 9) or 30 μg/kg of LPS, derived from *Escherichia coli* 055: B5 (Sigma-Aldrich, St Louis, MO), was administered intraperitoneally to mice, n = 9 each group. Three hours later, mice were performed the NOR test. The test consisted of the following three sessions (Fig 1B). 1) Habituation: Mice were allowed to habituate to an empty arena for 10 min. This session was used for the open field test. 2) Familiarization session: mice were allowed to explore two identical objects (plastic bottles covered with white glossy paper: familiar object) for 10 min. 3) Recognition session: after the familiarization session, one of the familiar objects was replaced by a new object (canister covered with black glossy paper: novel object), and the mice were allowed to explore the two objects for 10 min.

To avoid the effects of olfactory information, the arena interior and objects were wiped with 70% ethanol when the mice were removed from the arena between sessions. All experiments were carried out under 20–25 lux lighting conditions, and animal behavior was video-recorded using a camera (BB-HCM715, Switch-S8PWR, Panasonic, Osaka, Japan) installed in the arena. Video recordings were analyzed using analysis software (BORIS, http://www.boris. unito.it/). Exploratory behavior was defined as touching the tip of the nose or paw on an object, and the following variables were defined:

F: exploratory time for familiar objects during the recognition session

N: exploratory time for novel objects during the recognition session

Discrimination index: (N-F)/(N+F)

The discrimination index, rather than N or N-F, was adopted for further analysis in this study to cancel out variability in the total exploratory time of individual mice.

### Open field test

To evaluate the locomotor activity of the mice, a record of the 10 min habituation session at the beginning of the novel object recognition test was used as the open field test (Fig 1B). The mice were placed in the arena (40 cm x 40 cm) and allowed to explore for 10 min. The behavior was video-recorded, converted to an appropriate frame rate (2 fps) for analysis (Convertio, https://convertio.co/ja/), and then transferred to automatic analysis software (Time FZ1, O'HARA, Tokyo, Japan). The total traveling distance for each mouse was used as an indicator of locomotor activity. The time spent in the center area (20 cm x 20 cm) of the arena, and the number of entries into the same area were used as indicators for anxiety [30–32].

### Serum sample for IL-6 assay

Cardiac blood sampling and brain slice preparation were performed after the NOR test. Blood was sampled from the right ventricle of isoflurane- inhalation anesthetized mice before subsequent brain slice preparation. The collected blood was immediately dispensed into tubes

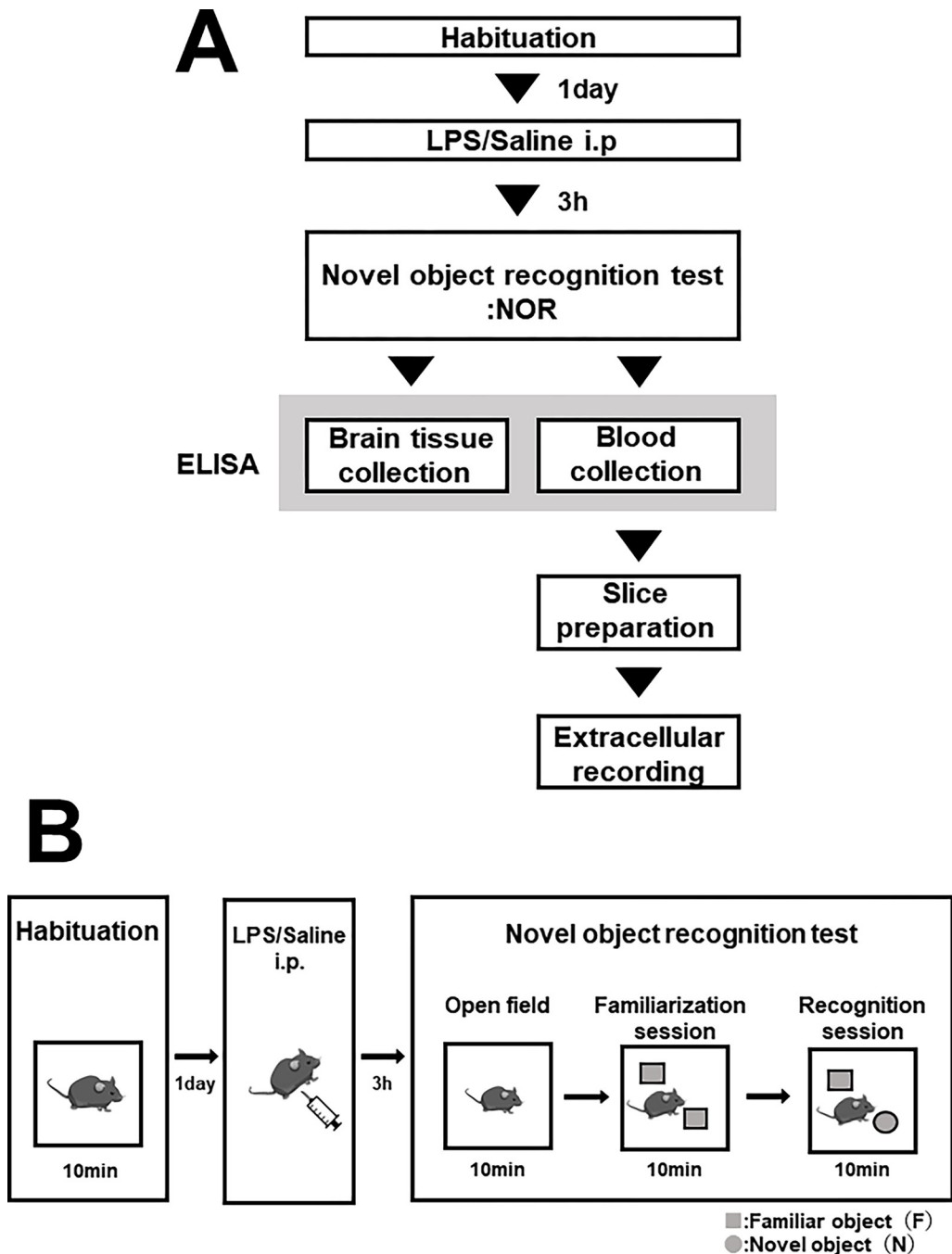

**Fig 1. Timeline of the experimental procedure.** (A) Diagram of the experimental procedure. (B) Diagram of novel object recognition test.

(Sepharabit Tube S; Tokuyama Sekisui, Yamaguchi, Japan). Blood samples were allowed to stand at room temperature for 30 min and then centrifuged at 1500 × g for 10 min to prepare serum samples (control group, n = 8; LPS-treated group, n = 9). Serum samples were stored at -80°C until enzyme-linked immunosorbent assay (ELISA).

## Brain tissue sample for IL-6 assay

To measure IL-6 concentrations in the cerebral cortex containing the medial prefrontal cortex (mPFC), we used a separate group of mice (Fig 1A). Mice (control group: n = 6, LPS group: n = 7) were intraperitoneally injected with either 0.9% saline or 30 μg/kg LPS, and the brains were excised using the identical procedure as slice preparation for electrophysiology. To prepare samples containing ACC, the olfactory bulb and cerebellum were removed, the brain was sliced coronally, and the brainstem was trimmed. The brain samples, weighing between 53.5 and 93.5 mg per mice, were homogenized in 1 mL of PBS buffer (in mM; NaCl 137, $Na_2HPO_4$ 8.1, KCl 2.68, $KH_2PO_4$ 1.47) containing protease inhibitors (Nacalai, Kyoto, Japan). After homogenization, the samples were centrifuged at 13,000 rpm for 20 min at 4°C. The resulting mPFC lysate supernatant was collected and frozen at -80°C until the quantification of IL-6.

## IL-6 assay

To assess the effect of LPS administration on peripheral and brain cytokines, we quantified the levels of the pro-inflammatory cytokine IL-6 (Fig 1A). IL-6 concentration was measured by a sandwich ELISA using ELISA MAX™ Deluxe Set Mouse IL-6 (Biolegend, San diego, CA) in a 96 well-plate (Nunc-Immuno Module plate F16, Thermo Fisher Scientific, Waltham, MA). The serum samples were diluted 100-fold for adjustment within the dynamic range of the calibration curve. The assay was performed according to manufacturer's instructions.

## Brain slice preparation and electrophysiology

Brain slices were prepared as described previously [27]. After cardiac blood collection, the mice were transcardially perfused with cooled cutting solution (in mM: choline chloride 120, KCl 3, $NaHCO_3$ 28, $NaH_2PO_4$ 1.25, glucose 22, and $MgCl_2$ 8) from the left ventricle, and their brains were rapidly excised and soaked in ice-cold cutting solution. Brains were placed on a vibrating-blade microtome (VT1000S; Leica, Wetzlar, Germany). Two or three 450 μm thick coronal brain slices (450 μm) containing the ACC were prepared from the brain submerged in ice-cold cutting solution saturated with 95% $O_2$ and 5% $CO_2$. The prepared slices were trimmed on both sides, placed on a film sheet (Nucleopore®, CORNING, Corning, NY), and transferred to a chamber filled with $O_2/CO_2$-saturated artificial cerebrospinal fluid (ACSF, in mM: NaCl 120, KCl 3, $CaCl_2$ 2.5, $MgCl_2$ 1.3, $NaHCO_3$ 26, $NaH_2PO_4$ 1.25, glucose 15). The brain slices were incubated in the chamber for at least 1 h before electrophysiology experiments.

Thereafter, the brain slice attached to the film sheet was transferred to a submerged recording chamber, and the slice was glued with agar at the four corners of the film to the bottom of the chamber. The slices were then perfused with $O_2/CO_2$-saturated ACSF at a rate of 6mL/ min. The solution was maintained at 26–28°C by a controller (TC-324B, WARNER Instrument, Holliston, MA).

Extracellular recordings were performed to obtain field potentials from the Cg1 region of the right ACC. Glass electrodes were made using a micropipette puller (P-97, SUTTER, Novato, CA, USA) and filled with NaCl (0.5 M). Electrodes were placed at the recording points: the superficial layer (layer II/III) of the dorsal-midline corner in the Cg1 region of the ACC and at approximately 150 μm inside the brain surface, as estimated under a microscope (Fig 2A).

To induce network oscillation *in vitro*, KA (K0250, Sigma-Aldrich, St. Louis, MO) was perfused for 5 min at 0.3, 1.0, and 3.0 μM concentrations in series (control group: n = 8, LPS-treated group: n = 9). To investigate the effect of histamine on KA-induced network oscillation, 10 μM histamine (H-7250, Sigma-Aldrich St. Louis, MO) was first perfused for 10 min,

and then KA was added in the presence of histamine at the abovementioned concentrations (control group: n = 7, LPS-treated group: n = 10, Fig 2B). The signal was amplified by a differential amplifier (DAM80, WPI, Sarasota, FL) and acquired at a sampling rate of 10 kHz via an interface (Digidata 1440A, Molecular Devices, San Jose, CA) with a band-pass filter between 0.1 Hz and 1 kHz and recorded on a PC.

The 5-minute time window was adopted for the analysis of network oscillations, from the time when the corresponding concentration of KA perfusion reached the slice (1 min after the start of KA perfusion). The power spectrum density (PSD) was obtained by fast Fourier transform using Clampfit 11.1 software (Molecular Devices, San Jose, CA, USA). The oscillation power component within each frequency range (theta: 3–8 Hz, alpha: 8–12 Hz, beta: 12–30 Hz, low gamma: 30–50 Hz, high gamma: 50–80 Hz, total: 3–80 Hz) was calculated from the area under the PSD curve. In this study, the delta (0.5–3 Hz) range was excluded from the analysis because of its low power.

## Statistics

To test the normality of the data, except for the oscillation power, the Shapiro-Wilk test was performed. A p-value less than 0.05 was considered significant. We performed parametric tests when the data values satisfied normality ($p > 0.05$); otherwise, non-parametric tests were performed. Locomotor activity in the open field session was analyzed using the Mann-Whitney test, discrimination index of the NOR test with an unpaired t-test, serum IL-6 concentration with the Mann-Whitney test, and brain-supernatant IL-6 concentration with the unpaired t-test. Non-parametric data are shown as box plots with the median, box (25–75th percentile), and bar (5–95th percentile).

The effects of LPS and histamine on oscillatory power were analyzed using two-way ANOVA. When significant interactions were observed, multiple comparisons were performed using the Steel-Dwass test. Values are shown as mean±SEM.

Significance of the correlation between oscillation power and behavioral profiles (total entries and the time spent in the center area of the open field, and NOR task data) was tested using t-tests with Pearson's coefficient.

The following software was used to obtain statistical data: Excel®(Microsoft, Redmond, WA), KyPlot ver.5.0(Kyenslab, Tokyo, Japan), JMP® ver.16.0.0(SAS Institute, Cary, NC). A value of $p < 0.05$ was considered statistically significant(*$p < 0.05$, **$p < 0.01$).

## Results

### Effects of LPS on IL-6 concentration

LPS-treated mice showed a significant increase in IL-6 concentration in the serum (Fig 3A) (p = 0.0002) and an increasing trend in the supernatant of the brain tissue, *i.e.*, the cerebral cortex, including the ACC (Fig 3B) (p = 0.177).

### Network oscillation in the ACC slice recording

A serial application of incrementally increasing concentrations of KA (0.3, 1.0, 3.0 μM for 5 min each) to the ACC slices induced network oscillation, intensified with the increasing dose of KA as described previously (Fig 4A) [26]. Brain slices obtained from LPS-treated mice exhibited more intense network oscillations (Fig 4B). Bath application of histamine also showed potentiating effects on network oscillations in saline-treated control mice (Fig 4C), but this effect was not observed in brain slices from LPS-treated mice (Fig 4D).

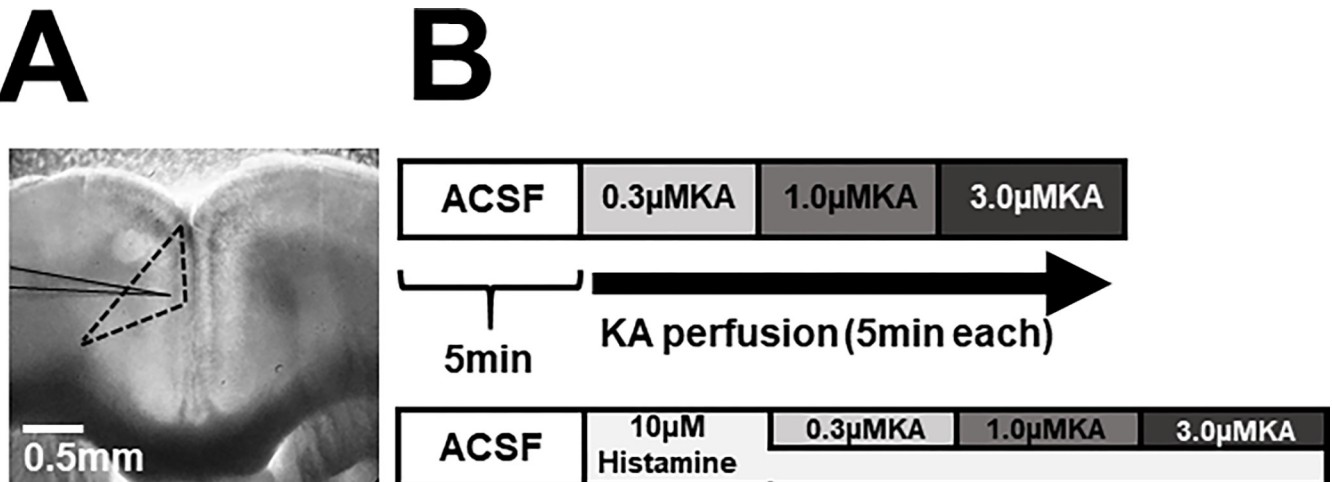

**Fig 2. Outline of electrophysiological recording.** (A) Microscopic image of the ACC. Extracellular field potentials were recorded from the superficial layer of the ACC slices. The black line indicates the position of the recording electrode. The dotted line indicates Cg1 region of ACC. (B) Timeline of drug perfusion protocol for extracellular recordings. Extracellular field potentials were recorded under the absence (top) or presence (bottom) of 10 μM histamine. Network oscillation was induced by KA perfusion. KA was perfused with increasing concentrations (0.3, 1.0, and 3.0 μM for 5 min each). Abbreviations: ACC, anterior cingulate cortex; KA, kainic acid.

## Effects of LPS on the oscillatory power

The augmentation of the oscillatory power by LPS pretreatment was indicated by ANOVA for the most frequency ranges except for high-gamma component, *i.e.*, theta $F_{(1, 30)} = 6.15$, $p = 0.019$), alpha $F_{(1, 30)} = 6.06$, $p = 0.020$), beta $F_{(1, 30)} = 4.29$, $p = 0.047$), low-gamma $F_{(1, 30)} = 4.27$, $p = 0.048$), and total frequency components $F_{(1, 30)} = 6.47$, $p = 0.016$), at the KA

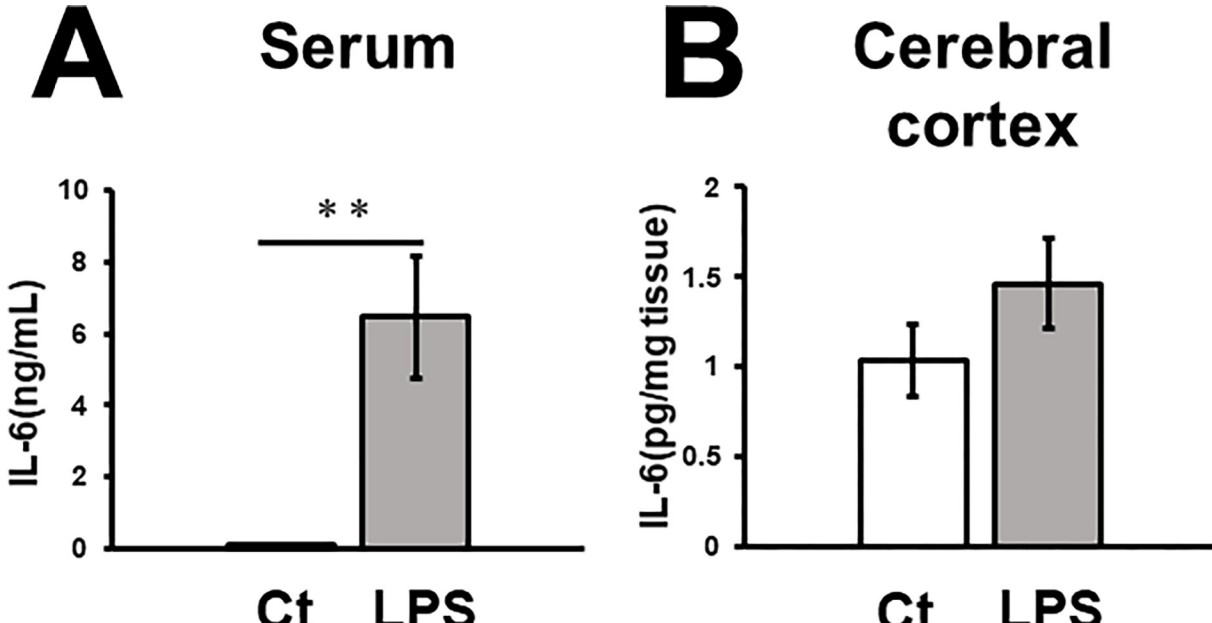

**Fig 3. In LPS-treated mice, IL-6 significantly increased in serum but not in the cerebral cortex.** The concentration of IL-6 in the serum (A) and cerebral cortex including ACC (B). The number of serum samples were as follows: control group: n = 8, LPS group: n = 9, **: $p < 0.01$, Mann-Whitney test. The number of the cortex samples: n = 6 for control group, and n = 7 for LPS group. Data are presented as mean±SEM.

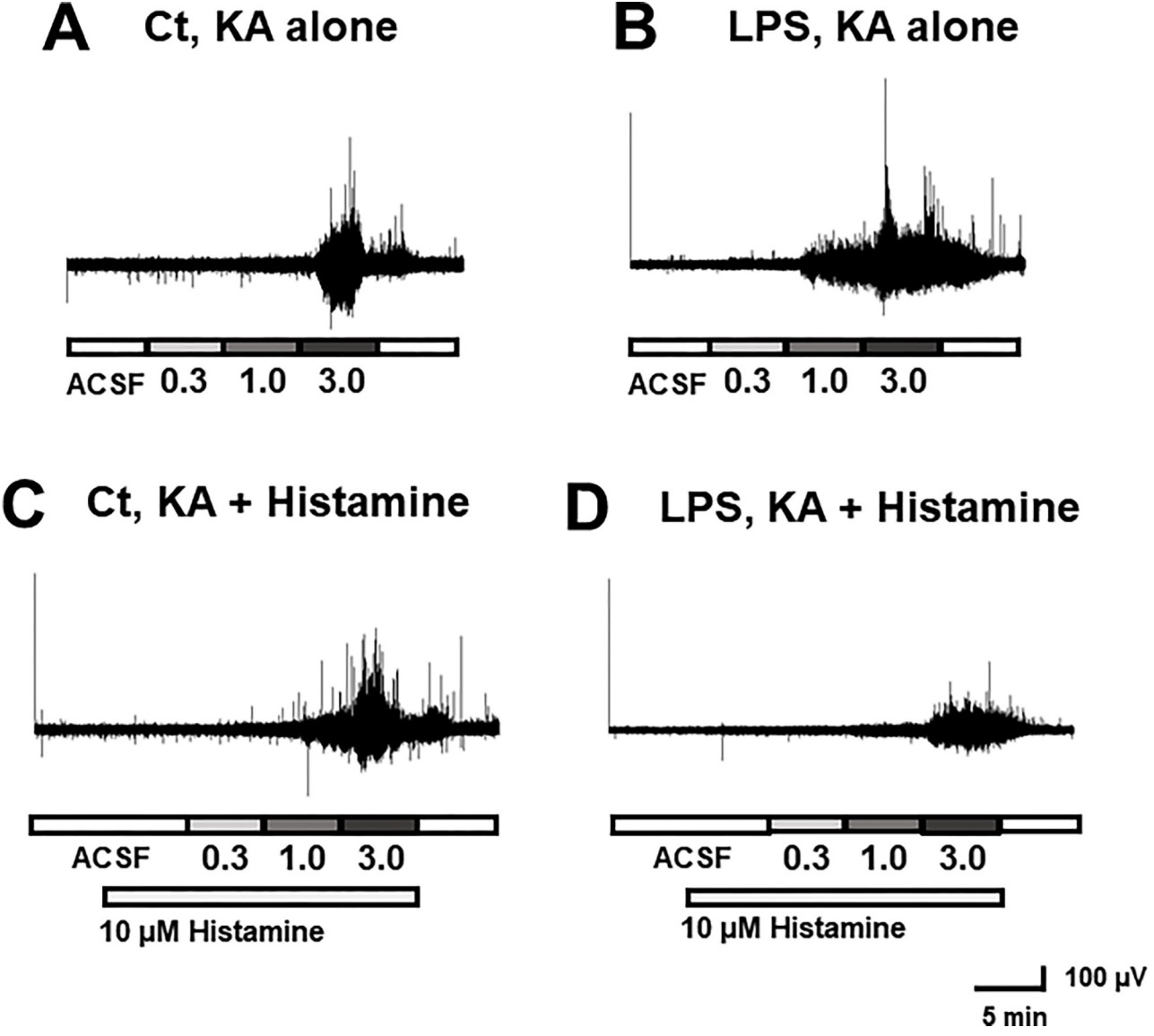

**Fig 4. The representative actions of LPS treatment and histamine application on the KA-induced network oscillation.** The numerical values 0.3, 1.0, and 3.0 indicate the KA concentrations in μM. Representative traces of KA-induced network oscillation from ACC slices of control mouse (A), LPS-treated mouse, KA (B), control mouse, KA+ histamine (C). LPS-treated-mouse, KA+ histamine (D).

concentration of 1.0 μM (Fig 5). The post-hoc comparisons revealed significant differences from LPS-treatment specifically for the alpha component (Fig 5B) (1.0 μM KA:4.5 ± 1.2 μV$^2$ for control, and 30.7 ± 8.9 μV$^2$ for LPS-treated mice).

## Effects of histamine on the oscillatory power

Potentiation of oscillatory power by histamine perfusion during KA-induced network oscillations was detected in the control group as a main effect using ANOVA (theta; 3.0 μM: $F_{(1, 30)}$ = 5.23, p = 0.030, alpha; 3.0 μM: $F_{(1, 30)}$ = 5.90, p = 0.021). Significant differences in the oscillatory power between KA-alone and KA plus histamine perfusion groups were observed in theta (Fig 5A) (3.0 μM KA: 28.4 ± 6.7 μV$^2$ for control and 77.2 ± 12.9 μV$^2$ for histamine

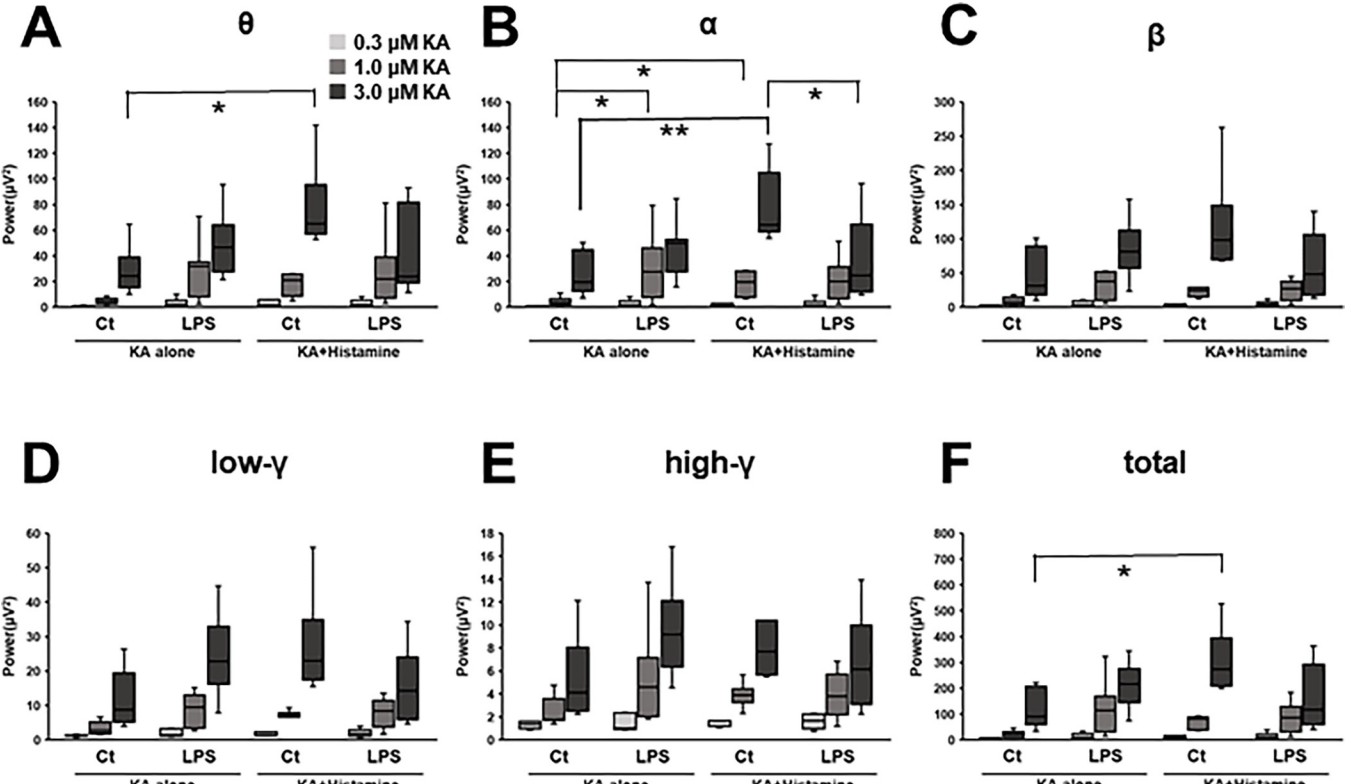

**Fig 5. The summarized effects of LPS treatment and histamine application on the KA-induced network oscillation.** (A) theta: 3–8Hz, (B) alpha: 8–12Hz, (C) beta: 12–30Hz, (D) low gamma: 30–50Hz, (E) high gamma: 50–80Hz, and the (F) total: 3–80Hz. *: $p < 0.05$, **: $p < 0.01$ by Steel-Dwass test. Two-way ANOVA showed the interaction between LPS and histamine in all frequency components. Steel-Dwass test was performed as a post-hoc test for each KA concentration. Control group (8 mice): 8 slices for KA alone, 7 slices for KA+ histamine. LPS group (9 mice): 9 slices for KA alone and 10 slices for KA + histamine. Abbreviations: LPS, lipopolysaccharide; KA, kainic acid.

perfusion), alpha (Fig 5B) (1.0 μM KA:$4.5 \pm 1.2$ μV$^2$ for control and $18.7 \pm 5.2$ μV$^2$ for histamine perfusion, and 3.0 μM KA:$26.5 \pm 6.5$ μV$^2$ for control and $76.2 \pm 11.5$ μV$^2$ for histamine perfusion), and total frequency components (Fig 5F) (3.0 μM KA: $120.9 \pm 28.5$ μV$^2$ for control and $307.3 \pm 48.0$ μV$^2$ for histamine perfusion.

### Interactive effects of LPS and histamine on the oscillatory power

In the LPS-treated group, histamine did not potentiate oscillatory activity, as indicated by the significant interaction between LPS treatment and histamine perfusion by 2-way ANOVA (theta; 3.0μM: $F_{(1, 30)} = 8.55$, $p = 0.007$, alpha; 1.0μM: $F_{(1, 30)} = 4.77$, $p = 0.037$, 3.0μM: $F_{(1, 30)} = 10.82$, $p = 0.003$, beta; 3.0μM: $F_{(1, 30)} = 7.32$, $p = 0.011$, low-gamma; 3.0μM: $F_{(1, 30)} = 9.60$, $p = 0.004$, high-gamma; 3.0μM: $F_{(1, 30)} = 5.63$, $p = 0.024$, total frequency components; 3.0μM: $F_{(1, 30)} = 11.19$, $p = 0.002$). Notably, oscillatory power was smaller for the LPS-treated group with histamine perfusion than that for the control group with histamine perfusion in the alpha component: $76.2 \pm 11.5$ μV$^2$ for control and $38.2 \pm 10.5$ μV$^2$ for LPS-treatment at 3.0 μM KA plus histamine perfusion (Fig 5B).

### Effect of LPS on behavior

Spontaneous locomotor activity evaluated by recordings during the open field session showed no differences after LPS treatment (Fig 6A) ($p = 0.321$). The number of entries into the center

area and the time spent in the same area during the open field session were adopted as indications for anxiety, but not to find statistical differences by LPS treatment when control and LPS-treated group were compared (Fig 6B and 6C) (p = 0.092, 0.729), respectively. The NOR test, an evaluation of cognitive performance, did not show significant differences in the discrimination index between the control and LPS-treated groups (Fig 6D) (p = 0.596).

Further analysis attempting to correlate oscillation power of the ACC field potential and behavioral parameters in the open field and NOR tests revealed that oscillation powers of theta (3–8 Hz) and total, and low-gamma (30–50 Hz) and high-gamma (50–80 Hz) bands correlated with total entries in the center area of the open field at 0.3 μM, and 3.0 μM KA, respectively (Table 1 for the first 5-minute data, shaded cells; S1 Table for 10-minute data). No statistical significance was observed in Pearson's coefficients between oscillation powers and the total time spent in the center area during the open field test or NOR test results. We combined two groups of mice shown in Fig 6 for this analysis.

## Discussion

In this study, we found that 30 μg/kg LPS administration to mice altered KA-induced network oscillation within the Cg1 region of the ACC *in vitro*, accompanied by increased IL-6 concentration in serum within 3–4 hours. The oscillatory power was increased by either LPS pretreatment or histamine application along with KA activation, whereas combined treatment LPS and histamine abolished this singular effect. KA-induced oscillatory power correlated with the total number of entries into the center area during the open field test. We did not observe any significant changes in the cognitive performance index by NOR test.

### Acute LPS actions on cognition, neural network oscillation, and cytokine production

LPS-induced systemic inflammation acutely induces sickness behavior with persisting disturbed cognitive function [35, 36]. Considering the potential involvement of the ACC in attention and arousal, possibly through its network oscillations, changes in the oscillatory activity are expected to affect cognitive functions and behaviors. Previous reports discuss the effects of LPS on cognitive functions mostly focusing at the time-frame 24 hours after LPS administration or later [16, 17, 37, 38]. Typical dosage of LPS in such studies are 0.1–5 mg/kg body weight, which could significantly reduce locomotor activity at earlier time point, e.g. 3 hours, after LPS-administration [15, 39, 40]. Since locomotor activity is often included in the readout of cognitive function tests, acute effects of LPS on cognitive function are something difficult to examine. The low dose of LPS used in this study (30 μg/kg body weight) enhanced the power of network oscillations in the ACC (Fig 5), but did not cause significant changes locomotor activity (Fig 6). Notably, the power of network oscillation was found correlated with an anxiety-related readout of the open field test for some frequency ranges (Table 1). In the NOR test, on the other hand, such correlations were not observed. We conclude that the low dose of LPS used in this study is suited for examining the acute effect of LPS on cognitive functions. Causal relationship between network oscillation and behavior would be clarified by further studies.

In the current study, IL-6 concentration increased in the serum 3–4 hours after LPS (30 μg/kg) administration, but this increase was not evident in the cerebral cortex. It appears consistent with the previous study, showing a minimal inducibility of IL-6 mRNA in the cortex 2 and 4 hours after a comparable dose of LPS (20 μg/kg) [41]. When higher dose LPS (e.g. 5 mg/kg) was administered, an acute IL-6 mRNA induction was observed robustly [42]. Microglia could produce IL-6 in response to LPS and inflammatory cytokines; however, typical readouts for microglia activation, such as ramification and density alteration are not evident in the

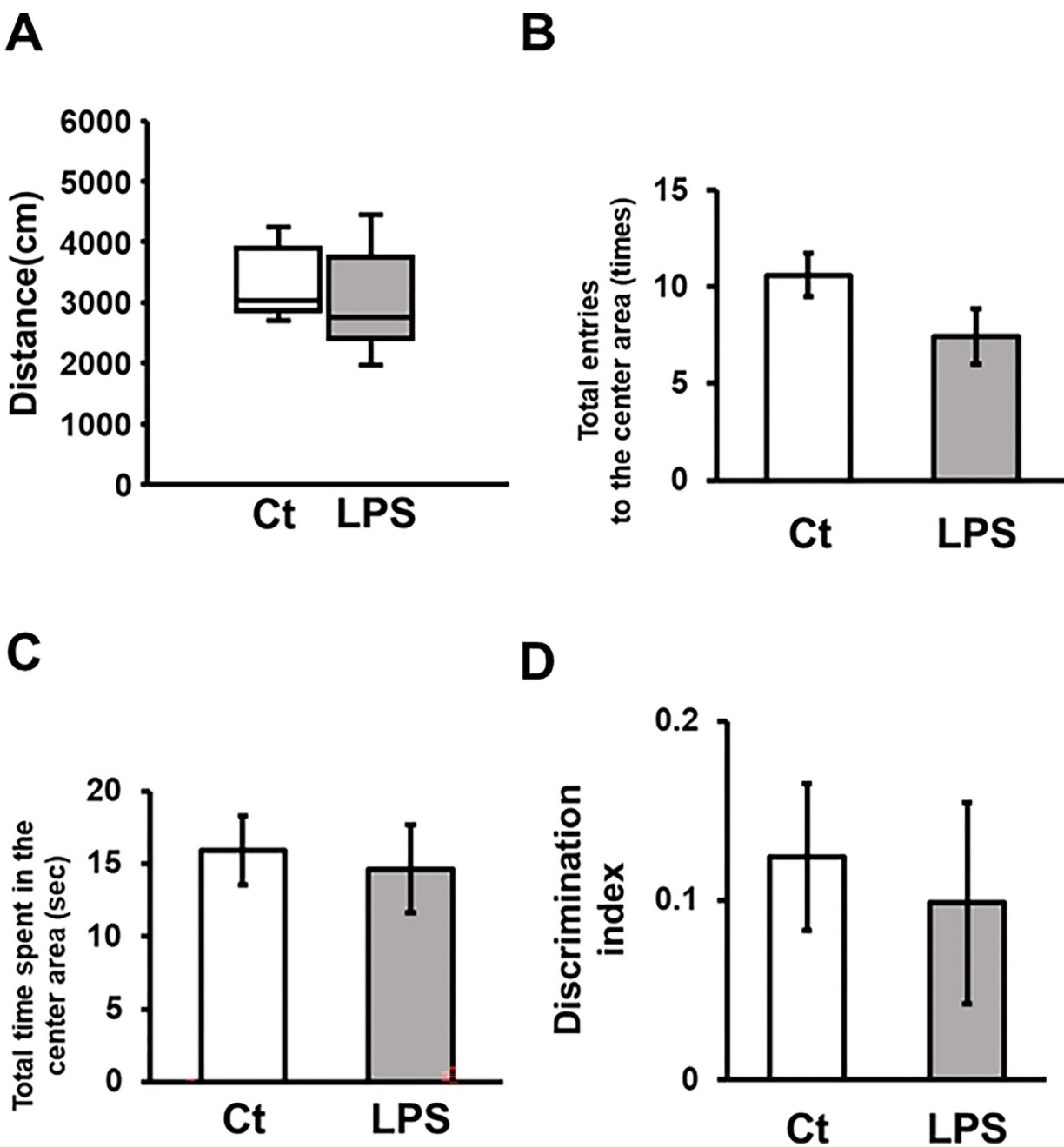

**Fig 6. Locomotor activity and cognitive functions were unaltered by LPS administration.** (A) Total distance traveled in the open field during the 10-minute recording period was analyzed by Mann-Whitney test. (B, C) The total number of entries into the open field center area, and the total time spent in the same area in the first 5 minutes were analyzed by an unpaired t-test. (D) Discrimination index on NOR test was analyzed with an unpaired t-test (control group: n = 8, LPS group: n = 9).

earlier time frame [43]. Involvement of microglia in the acute phase responses after low dose LPS administration will be a focus of the further study. We did not find a correlation between IL-6 production and the power of the ACC network oscillations from our current observations.

**Table 1. Correlation between KA-induced oscillation powers and behavioral parameters (n = 17)[1]).**

| Band | KA conc.(μM) | vs total entries to the center area[2]) | | vs total time spent in the center area[2]) | | vs discrimination index | |
|---|---|---|---|---|---|---|---|
| | | r [3]) | p value [4]) | r | p value | r | p value |
| theta | 0.3 | -0.522 | 0.032 | -0.177 | 0.496 | -0.095 | 0.717 |
| | 1.0 | -0.289 | 0.260 | 0.283 | 0.271 | -0.106 | 0.687 |
| | 3.0 | -0.071 | 0.787 | 0.386 | 0.126 | -0.237 | 0.359 |
| alpha | 0.3 | -0.457 | 0.065 | -0.146 | 0.576 | -0.060 | 0.818 |
| | 1.0 | -0.171 | 0.511 | 0.189 | 0.468 | -0.265 | 0.305 |
| | 3.0 | -0.248 | 0.337 | 0.165 | 0.527 | -0.295 | 0.250 |
| beta | 0.3 | -0.419 | 0.094 | -0.090 | 0.732 | 0.049 | 0.853 |
| | 1.0 | -0.169 | 0.515 | 0.035 | 0.894 | -0.327 | 0.201 |
| | 3.0 | -0.446 | 0.073 | -0.183 | 0.483 | -0.125 | 0.632 |
| low gamma | 0.3 | -0.343 | 0.177 | -0.065 | 0.803 | -0.164 | 0.528 |
| | 1.0 | -0.181 | 0.488 | -0.022 | 0.933 | -0.348 | 0.171 |
| | 3.0 | -0.560 | 0.019 | -0.327 | 0.200 | -0.215 | 0.407 |
| high gamma | 0.3 | -0.203 | 0.434 | 0.048 | 0.856 | -0.090 | 0.731 |
| | 1.0 | -0.212 | 0.413 | -0.009 | 0.974 | -0.344 | 0.177 |
| | 3.0 | -0.571 | 0.017 | -0.282 | 0.272 | -0.188 | 0.469 |
| total | 0.3 | -0.484 | 0.049 | -0.128 | 0.624 | -0.044 | 0.868 |
| | 1.0 | -0.212 | 0.414 | 0.126 | 0.629 | -0.289 | 0.260 |
| | 3.0 | -0.376 | 0.137 | -0.002 | 0.993 | -0.221 | 0.394 |

1) Two groups of mice as described in Fig 6 were combined, Control (n = 8), LPS (n = 9). 2) The first 5-minute data of the open field test were used for total entries to the center area and total time spent in the center area [33, 34]. 3) Correlations assessed by Pearson's coefficient. 4) t-tests with Pearson's coefficient.

## Effects of histamine on oscillation activity

When co-applied with KA, histamine increased the oscillatory power of the ACC network induced by KA in the control group. Considering the sedating effect of H1 antihistamine drugs, this change in oscillatory power appears to be related to cognitive processing in a more straightforward fashion through the H1 receptor than the other histamine subtypes. Andersson *et al*. reported a decremental function of the histamine H3 receptor in KA-induced hippocampal gamma oscillations [44], whereas enhancement of activity in theta and gamma oscillations was indicated in the medial entorhinal cortex [45]. Each histamine receptor subtype may play a distinct role in network oscillations in different areas of the CNS.

The distribution of histamine receptors in the mouse ACC and changes in their expression during acute inflammation are not well documented. Functionally, H1 receptor expressed in ACC is reported to be involved in depressive disorders [46], and H2 and H3 receptors to play roles in nociception [47] by animal model studies with receptor type-specific antagonists. But, the relationship between histamine-involved behavior and network oscillation in the ACC has not been well investigated. The delineation of histaminergic neurons in the neuronal circuit of the ACC is the key to clarifying the effects of LPS or inflammation on ACC function.

Oscillations in the delta range are not discussed in this study because of their low power in our experimental settings. Delta range oscillations are reportedly related to shallow sleep and impaired wakefulness [48]. Analysis of this frequency range will be the focus of future research.

## Interaction between LPS and histamine

Perfusion of histamine to the brain slices obtained from LPS-treated mice resulted in neuronal network oscillation with properties distinct from those of the histamine-perfused slices from control mice and non-histamine-perfused slices from LPS-treated mice. Receptors for LPS and histamine exert their functions through distinct cell biological pathways. Still, ANOVA results pointed out a significant interaction between LPS treatment and histamine perfusion. This finding indicated that LPS inhibited the oscillation-enhancing effect of histamine or that histamine counteracted the oscillation-promoting effect of LPS. However, the relationship between LPS and histamine remains unclear. LPS administration increases CNS histamine concentrations [25], suggesting that excessive histamine release downregulates histamine receptors, diminishing the oscillation-enhancing effect of histamine. Because subtype-specific histaminergic agonists or antagonists were not used in this study, we could not identify the histamine receptor subtype responsible for our findings.

If the LPS-IL-6 system suppresses histamine action, the histaminergic system during inflammation may differ from that in a non-inflammatory state. Further molecular biological analyses combined with behavioral, electrophysiological, and pharmacological approaches are necessary to clarify the interactions between inflammatory cytokines and the histaminergic system. Experiments using ACC slice preparations from LPS-pre-treated mice have a distinct advantage for this purpose. The mechanisms underlying the inhibitory interaction of LPS and histamine should be further clarified in the future.

In summary, our study revealed that acute inflammation induced by LPS enhanced network oscillation in the Cg1 region of the ACC, for which correlations were shown for certain animal behavior. We also observed that histamine enhanced network oscillations in non-LPS-treated mice. Unexpectedly, this enhancement was counteracted by LPS pretreatment. An increased concentration of serum IL-6 was observed in LPS-treated mice; however, this change was not observed in the cerebral cortex. Further studies are needed to clarify which cytokines are responsible for the LPS-induced effects on network oscillations and behavioral changes.

## Supporting information

**S1 Table. Correlation between KA-induced oscillation powers and behavioral parameters.** In this table, 10-minute data of the open field test were used for total entries to the center area and total time spent in the center area.
(XLSX)

**S1 Dataset. Raw data of this study.**
(XLSX)

## Acknowledgments

We are grateful to the animal facility members of Saitama Medical University for their assistance with animal care and Dr. Takanari Nakano for critical advice.

## Author Contributions

**Conceptualization:** Takayuki Murakoshi, Naonori Uozumi.

**Data curation:** Ayumi Hirao.

**Formal analysis:** Ayumi Hirao, Yasushi Hojo.

**Funding acquisition:** Takayuki Murakoshi.

**Investigation:** Ayumi Hirao, Yasushi Hojo, Gen Murakami, Rina Ito, Miki Hashizume.

**Resources:** Ayumi Hirao, Gen Murakami.

**Supervision:** Yasushi Hojo, Takayuki Murakoshi, Naonori Uozumi.

**Writing – original draft:** Ayumi Hirao.

**Writing – review & editing:** Ayumi Hirao, Yasushi Hojo, Takayuki Murakoshi, Naonori Uozumi.

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
