## [Decision Letter · Decision Letter 0]

5 Jan 2024

PONE-D-23-39823Effects of systemic inflammation on the network oscillation in the anterior cingulate cortex and cognitive behaviorPLOS ONE

Dear Dr. Hojo,

Thank you for submitting your manuscript to PLOS ONE. After careful consideration, we feel that it has merit but does not fully meet PLOS ONE’s publication criteria as it currently stands. Therefore, we invite you to submit a revised version of the manuscript that addresses the points raised during the review process.

 **The two reviewers addressed a number of major and minor concerns about your manuscript. Please revise your manuscript according to the comments from reviewers.**

We look forward to receiving your revised manuscript.

Kind regards,

Kenji Hashimoto, PhD

Section Editor

PLOS ONE

Journal Requirements:

Reviewers' comments:

Reviewer's Responses to Questions

**Comments to the Author**

1. Is the manuscript technically sound, and do the data support the conclusions?

Reviewer #1: Partly

Reviewer #2: Yes

2. Has the statistical analysis been performed appropriately and rigorously? 

Reviewer #1: Yes

Reviewer #2: Yes

3. Have the authors made all data underlying the findings in their manuscript fully available?

Reviewer #1: Yes

Reviewer #2: Yes

4. Is the manuscript presented in an intelligible fashion and written in standard English?

Reviewer #1: Yes

Reviewer #2: Yes

5. Review Comments to the Author

Reviewer #1: In this study, authors investigated the effects of systemic inflammation on ACC network oscillation on cognitive function, and obtained the results that 1) LPS increased network oscillatory power in the ACC Cg1 region, 2) LPS increased IL-6 in periphery but not in the cerebral cortex, 3) histamine increased network oscillation, which was suppressed by LPS, 4) LPS treatment did not affect cognitive function measured by NOR test. Based on these findings, authors concluded that the present experimental setting has a distinct advantage in obtaining mechanistic insights into inflammatory cognitive impairment. Although the objective of the present study is important and interesting, data obtained in the present study are rather complicated, and there are too many speculations, which make authors conclusion more ambiguous. Therefore, authors need to massively rewrite discussion or conduct additional experiments in which cognitive dysfunction is observed.

1. The biggest problem of the present study is too many speculations on behavioral results (cognitive dysfunction). Authors raised many reasons why authors could not detect cognitive impairment in the NOR test in the present condition, which raises the possibility that the present condition of the behavioral study is not appropriate and the present behavioral study is rather preliminary. If authors would like to stick to the original hypothesis, authors must demonstrate with experimental results. Therefore, authors need to 1) totally change conclusion that LPS-induced alteration of network oscillation is not related to cognitive dysfunction and massively rewrite abstract and discussion or 2) find appropriate conditions (LPS dose and NOR test, etc.) to impair cognition in the NOR test and investigate again network oscillation at the same condition as the behavioral study.

2. The description that NOR test was conducted to evaluate arousal levels (line 101) is not appropriate, because it does not specifically measure arousal level and some compounds enhance object recognition at dose(s) unaffecting arousal level. Please rephrase it.

3. Discussion, line 309. The phase “These results suggest that LPS and histamine share a common mechanism for enhancing oscillation power” may not be appropriate, because combination of LPS and histamine rather decreases oscillatory power and if LPS and histamine share a common mechanism, they may enhance but not decrease the power. Moreover, if LPS down-regulates histamine receptor by excessive histamine release, as authors speculated, LPS may be able to impair cognition as well.

4. Did authors confirm that the present NOR condition is appropriate to investigate impairment of preference to a novel object. In case authors would change conclusion, it would be recommendable to confirm that impairment of object cognition can be detectable in the present condition.

5. I wonder if 3 hours after LPS treatment would be appropriate to measure cognition, because behavioral changes at 3 hours after LPS administration are regarded as alterations due to sickness behavior. Please provide the plausible reason why authors conducted studies at 3 hours (but not at 24 hours) after LPS treatment.

Reviewer #2: The study by Hirao et al. is an interesting study showing the impact of LPS treatment on network oscillations in the ACC using an in vitro/ex vivo model. These findings are novel and intriguing. However, there are several issues that need to be addressed to increase the impact and connect different data sets with each other more rigorously either with new experiments (points 1, 2 and 5), reanalysis (points 2, 3 and 4) or discussion (points 1 and 5):

1) Rather acute effects of LPS (3 hours) were tested in this study. It is not clear if recovery from acute LPS treatment (~72 h) would have differential impact on the behaviour and ACC oscillations? This is interesting because LPS treatment causes drastic changes in microglia abundance/phenotype which might not be apparent during this acute phase. In this regard, it would be ideal to test whether the changes reported by the authors are related with any alterations in microglia density in the ACC. This can be checked with simple microglia-specific immunohistochemistry (IBA-1). This potential issue can be added to discussion.

2) In the methods section, it is mentioned that the “N=3 mice” is used for assessing IL-6 concentration in the cortex. However, in the figure legend (Fig. 3) it is mentioned that “The number of the cortex samples: n = 6 for control group, and n = 7 for LPS group”. This discrepancy should be corrected. Having only 3 mice per group is not statistically robust enough. If this is the case, I suggest increase of the experimental group with a second batch of animals to increase the statistical validity of the results.

3) I also suggest using log power of oscillations due to variable nature of oscillation power in in vitro/ex vivo slice preparation. This might help with reducing the variability.

4) It is not clear how ACC oscillations are related with novel object recognition task? Are there studies that directly relate ACC activity with the performance in this task? Why not a simple elevated plus maze etc. where anxiety and risk assessment behaviour (related with ACC function) can be assessed? Alternatively, the time spent in the centre of open field (which was test in the current study) can be used as a proxy for anxiety. Did the authors also try to correlate the behavioural readouts with oscillatory readouts (power?)? This would give a nice piece of information connecting behaviour and oscillation data. These data can be added to Fig. 6.

5) The results showing the impact of histamine on ACC oscillations and its interaction with LPS-treatment are interesting. While authors discuss these findings nicely (although, in part, generic), in my opinion, they do not take advantage of the slice model fully to mechanistically study this interaction. For example, which histamine receptors are involved in the observed enhancing vs. suppressing effects in the control vs. LPS group. Are these changes related with altered expression of the receptors in the ACC?

6. PLOS authors have the option to publish the peer review history of their article (what does this mean?). If published, this will include your full peer review and any attached files.

Reviewer #1: No

Reviewer #2: **Yes: **Gürsel Caliskan

---

## [Author Response · Author response to Decision Letter 0]

22 Feb 2024

Dear Editors and Reviewers

Thank you very much for reviewing our manuscript and giving us invaluable comments. Please find our point-by-point responses in the following pages.

We reanalyzed the presented data following the reviewers’ comments to find correlations between KA-induced oscillatory power and the open-field test result. This led us to change our conclusions, and extensive revision of the manuscript. One table, Table 1, and two figure panels, Fig 6B and C, were added in the revised manuscript to describe the reanalysis result. 

In the revised manuscript, we presented our emphasis on acute phase responses, or sickness behavior, plainly answering to the comments by both reviewers.

During the revision process, we noticed a wrong p-value for Fig 3A data. Corrected p-value, p = 0.0002, is described in l.246, p.14. This correction does not affect conclusions, or discussions of the manuscript.

Reviewer #1:

1) The biggest problem of the present study is too many speculations on behavioral results (cognitive dysfunction). Authors raised many reasons why authors could not detect cognitive impairment in the NOR test in the present condition, which raises the possibility that the present condition of the behavioral study is not appropriate and the present behavioral study is rather preliminary. If authors would like to stick to the original hypothesis, authors must demonstrate with experimental results. Therefore, authors need to 1) totally change conclusion that LPS-induced alteration of network oscillation is not related to cognitive dysfunction and massively rewrite abstract and discussion or 2) find appropriate conditions (LPS dose and NOR test, etc.) to impair cognition in the NOR test and investigate again network oscillation at the same condition as the behavioral study.

Ans. 1)

We reanalyzed the data set for open field test to note statistical significance on cognitive functions, i.e. correlations between KA-induced oscillatory power and the total number of entries into the center area, as shown in Table 1, page 19. This changed the conclusion of the current study. Accordingly, we extensively revised Abstract, Introduction, Results, and Discussion sections, removing speculations regarding the NOR test result as many as possible.

2) The description that NOR test was conducted to evaluate arousal levels (line 101) is not appropriate, because it does not specifically measure arousal level and some compounds enhance object recognition at dose(s) unaffecting arousal level. Please rephrase it.

Ans. 2)

We corrected the description of NOR test in the Materials and Methods section that reads as follows: The novel object recognition test (NOR) was conducted as a test for memory, following the protocol by Jung et. al (28) and by Ennaceur et. al (29) with slight modifications (ll. 100,101).

3) Discussion, line 309. The phase “These results suggest that LPS and histamine share a common mechanism for enhancing oscillation power” may not be appropriate, because combination of LPS and histamine rather decreases oscillatory power and if LPS and histamine share a common mechanism, they may enhance but not decrease the power. Moreover, if LPS down-regulates histamine receptor by excessive histamine release, as authors speculated, LPS may be able to impair cognition as well.

Ans. 3)

We removed the sentence pointed out by the reviewer, and revised the following 2 sentences in the Discussion that reads: Receptors for LPS and histamine exert their functions through distinct cell biological pathways. Still, ANOVA results pointed out a significant interaction between LPS treatment and histamine perfusion (ll. 407-409).

While two receptors are distinct as for the intracellular signaling pathways, when we look at the neuronal network level, two signals are interacting with each other.

4) Did authors confirm that the present NOR condition is appropriate to investigate impairment of preference to a novel object. In case authors would change conclusion, it would be recommendable to confirm that impairment of object cognition can be detectable in the present condition.

Ans. 4)

Our experimental conditions of NOR test are essentially the same as the cited references (28 and 29). We believe that our NOR test was carried out appropriately.

A reanalysis revealed statistical significances for open field test as explained in the response to in the reviewer’s comment. Now, we totally revised the paragraph “acute LPS actions on cognition, neural network oscillation, and cytokine production” of discussion section, putting less emphasis on NOR test results(ll.351-378). We do not consider that the negative results for NOR test profoundly affected the conclusion of the current study.

5. I wonder if 3 hours after LPS treatment would be appropriate to measure cognition, because behavioral changes at 3 hours after LPS administration are regarded as alterations due to sickness behavior. Please provide the plausible reason why authors conducted studies at 3 hours (but not at 24 hours) after LPS treatment.

Ans. 5)

A major point of this study is that changes in the cognitive functions can be analyzed for the acute phase by choosing low dose LPS.

As mentioned by the reviewer, cognitive functions in the inflammatory state were more often discussed during the later time points (24 hr or later). This does not mean that cognitive changes only take place in the chronic phase. Rather, cognitive changes do occur in the first couple of hours after LPS administration, and are not readily detectable in animal models due to the more profound locomotor dysfunction especially when a high dose of LPS (~ mg/kg) are adopted.

We took advantage of choosing a low dose LPS (30 µg/kg), to minimize the effects on locomotor system during the acute phase, and demonstrated correlations between the cognitive function and the neuronal network oscillation as explained in the response to comment #1 of the reviewer, and in Table 1 in the revised manuscript.

We accordingly revised the paragraph “Acute LPS actions on cognition, neural network oscillation, and cytokine production” in the Discussion extensively.

 

Reviewer #2: 

1) Rather acute effects of LPS (3 hours) were tested in this study. It is not clear if recovery from acute LPS treatment (~72 h) would have differential impact on the behaviour and ACC oscillations? This is interesting because LPS treatment causes drastic changes in microglia abundance/phenotype which might not be apparent during this acute phase. In this regard, it would be ideal to test whether the changes reported by the authors are related with any alterations in microglia density in the ACC. This can be checked with simple microglia-specific immunohistochemistry (IBA-1). This potential issue can be added to discussion.

Ans. 1) 

Cognitive functions in the inflammatory state were more often discussed during the later time points (24 hr or later). This does not mean that cognitive changes only take place in the chronic phase. Rather, cognitive changes do occur in the first couple of hours after LPS administration, and are not readily detectable in animal models due to the more profound locomotor dysfunction especially when a high dose of LPS (~ mg/kg) are adopted.

Reanalysis of our open field test data demonstrated correlations between the cognitive function and the neuronal network oscillation, as described in the response to comment #4 of the reviewer. Thus, changes in the cognitive functions can be analyzed as an acute phase response to low dose LPS (30 µg/kg). It is of interest how this low-dose LPS effect in the cognitive functions lasts or changes in the later time points, which is a target for the coming study.

Discussion on microglia is added to the revised text (ll. 373-378) that reads: Microglia could produce IL-6 in response to LPS and inflammatory cytokines, however, typical readouts for microglia activation, such as ramification and density alteration are not evident in the earlier time frame (41). Involvement of microglia in the acute phase responses after low dose LPS administration will be a focus of the further study.

2) In the methods section, it is mentioned that the “N=3 mice” is used for assessing IL-6 concentration in the cortex. However, in the figure legend (Fig 3) it is mentioned that “The number of the cortex samples: n = 6 for control group, and n = 7 for LPS group”. This discrepancy should be corrected. Having only 3 mice per group is not statistically robust enough. If this is the case, I suggest increase of the experimental group with a second batch of animals to increase the statistical validity of the results.

Ans. 2) 

We thank the reviewer to point out our mistake. The wrong description in the Materials and Methods section in the initial submission was corrected (ll. 153,154) to be consistent with the legend for Fig 3.

3) I also suggest using log power of oscillations due to variable nature of oscillation power in in vitro/ex vivo slice preparation. This might help with reducing the variability.

Ans. 3)

We carried out the whole set of reanalysis with log power of oscillation data, but to obtain the essentially the same result as the original analysis using linear-scale data. We appreciate the reviewer’s suggestion, and present the same set of data as the initial submission.

4) It is not clear how ACC oscillations are related with novel object recognition task? Are there studies that directly relate ACC activity with the performance in this task? Why not a simple elevated plus maze etc. where anxiety and risk assessment behaviour (related with ACC function) can be assessed? Alternatively, the time spent in the centre of open field (which was test in the current study) can be used as a proxy for anxiety. Did the authors also try to correlate the behavioural readouts with oscillatory readouts (power?)? This would give a nice piece of information connecting behaviour and oscillation data. These data can be added to Fig 6.

Ans. 4）

 Reference 2, which investigated the relation between ACC (prefrontal cortex) and novel object recognition test, was already cited in the Introduction section of the initial submission. Since we put more emphasis on open field test results in the revised manuscript, as mentioned below, discussion on the potential roles of ACC in novel object recognition was cut short with an extensive revision of the paragraph "Acute LPS actions on cognition, neural network oscillation, and cytokine production” in the Discussion section.

We reanalyzed the data set for open field test. We noted statistical significance on cognitive functions, i.e. correlations between KA-induced oscillatory power and the total number of entries into the center area, as shown in Table 1, page 19. Simple comparisons between control and LPS-treated groups were not statistically significant (Fig 6B and C). Accordingly, we extensively revised the Abstract, Introduction, Results, and Discussion sections, changing the conclusion of the current study. We deeply appreciate the reviewer’s suggestion.

5) The results showing the impact of histamine on ACC oscillations and its interaction with LPS-administration are interesting. While authors discuss these findings nicely (although, in part, generic), in my opinion, they do not take advantage of the slice model fully to mechanistically study this interaction. For example, which histamine receptors are involved in the observed enhancing vs. suppressing effects in the control vs. LPS group. Are these changes related with altered expression of the receptors in the ACC?

Ans. 5) 

We understand that elucidation of the histamine receptor subtype and the cell type which express the relevant receptor is the key to clarify the mechanism underlying the role of histamine signaling in the CNS during LPS-induced inflammatory state. We are now preparing the brain slice electrophysiology experiments using histamine receptor antagonists for the further study. We mentioned the reported roles of histamine receptors in ACC function citing 2 more references in the paragraph “Effects of histamine on oscillation activity” in the Discussion section taking in the points suggested by the reviewer (ll. 390-397).

---

## [Decision Letter · Decision Letter 1]

6 Mar 2024

PONE-D-23-39823R1Effects of systemic inflammation on the network oscillation in the anterior cingulate cortex and cognitive behaviorPLOS ONE

Dear Dr. Hojo,

Thank you for submitting your manuscript to PLOS ONE. After careful consideration, we feel that it has merit but does not fully meet PLOS ONE’s publication criteria as it currently stands. Therefore, we invite you to submit a revised version of the manuscript that addresses the points raised during the review process.

We look forward to receiving your revised manuscript.

Kind regards,

Kenji Hashimoto, PhD

Section Editor

PLOS ONE

Journal Requirements:

**Additional Editor Comments:**

The reviewer#2 has several minor concerns about your revised manuscript. Please revise your manuscript carefully.

Reviewers' comments:

Reviewer's Responses to Questions

**Comments to the Author**

1. If the authors have adequately addressed your comments raised in a previous round of review and you feel that this manuscript is now acceptable for publication, you may indicate that here to bypass the “Comments to the Author” section, enter your conflict of interest statement in the “Confidential to Editor” section, and submit your "Accept" recommendation.

Reviewer #1: All comments have been addressed

Reviewer #2: (No Response)

2. Is the manuscript technically sound, and do the data support the conclusions?

Reviewer #1: Yes

Reviewer #2: (No Response)

3. Has the statistical analysis been performed appropriately and rigorously? 

Reviewer #1: Yes

Reviewer #2: (No Response)

4. Have the authors made all data underlying the findings in their manuscript fully available?

Reviewer #1: Yes

Reviewer #2: (No Response)

5. Is the manuscript presented in an intelligible fashion and written in standard English?

Reviewer #1: Yes

Reviewer #2: Yes

6. Review Comments to the Author

Reviewer #1: Authors responded adequately to my comments, and revised the manuscript accordingly. I appreciare authors' efforts to improve the manuscript. I have no further comments.

Reviewer #2: I strongly encourage to tone down the statements involving causal involvement of ACC oscillations in behavioural readouts throughout the manuscript. This is also true for the involvement of Histamine as a mediator of the reported effects. First, there is only a minor correlation with a behavioural parameter in the open field test. Second, the oscillations recorded in this study are ex vivo/in vitro and their causal involvement in behavioural readouts are of correlative nature without any causal evidence at this stage.

-Page 6, Line 84-85: rephrase this sentence as these are two different tasks assessing anxiety/activity and memory, respectively:

”…open field and novel object recognition, a behavioral task 84 related to ACC function (2,6)….”

-Page 8, Line137-139 and Page 18, Line314-318: It is not clear to me how the parameters assessed in the open field test are related to motivation.

-I suggest dividing the open field test data into 5 min bins as the anxiety/activity measures are more reliable at the beginning of this test.

-Regarding the correlation between the behavioural parameters and oscillation power. It would be interesting to do correlation separately within control and LPS group as the direction of the correlation can change in control vs LPS.

-For the correlation, the time spent in the centre is not reported. I think this is a better readout then the entries. Also, the correlation should be performed with the data corresponding to the first 5 min of the open field as mentioned above.

There are still some minor grammatical issues that can be corrected during the proofreading stage.

7. PLOS authors have the option to publish the peer review history of their article (what does this mean?). If published, this will include your full peer review and any attached files.

Reviewer #1: No

Reviewer #2: No

---

## [Author Response · Author response to Decision Letter 1]

2 Apr 2024

Dear Editors and Reviewers,

Thank you very much for reviewing our manuscript and giving us invaluable comments.

Please find our point-by-point responses in the following pages.

We reanalyzed our data following the reviewer 2’s comments. As a consequence, Table 1 and Fig 6B and C were replaced with the new ones, and original Table 1 is presented as Supplemental Table 1.

Conclusions of the manuscript is preserved after the reanalysis, and several changes were made in the text to meet the suggestions by the reviewer 2.

Reviewer #2: I strongly encourage to tone down the statements involving causal involvement of ACC oscillations in behavioural readouts throughout the manuscript. This is also true for the involvement of Histamine as a mediator of the reported effects. First, there is only a minor correlation with a behavioural parameter in the open field test. Second, the oscillations recorded in this study are ex vivo/in vitro and their causal involvement in behavioural readouts are of correlative nature without any causal evidence at this stage.

Ans.)　

We made changes in the manuscript following the suggestion by the reviewer to tone down the statements regarding causality or correlation. The revised sentence in the abstract, ll. 32-36, reads: These results suggest that LPS-induced systemic inflammation results in increased network oscillation and a drastic change in histamine sensitivity in the ACC, accompanied by the robust production of systemic pro-inflammatory cytokines in the periphery, and that these alterations in the network oscillation and animal behavior as an acute phase reaction relate with each other. We add a sentence in the discussion, ll.371-372, that reads: Causal relationship between network oscillation and behavior would be clarified by further studies.

Q1

-Page 6, Line 84-85: rephrase this sentence as these are two different tasks assessing anxiety/activity and memory, respectively:

”…open field and novel object recognition, a behavioral task 84 related to ACC function (2,6)….”

Ans. 1) 

We revised the sentence as suggested by the reviewer. The revised sentence reads: We analyzed the effects of acute LPS administration on 1) kainic acid (KA)-induced network oscillation from the Cg1 region of the ACC, either in the presence or absence of histamine; 2) anxiety by open field (6); 3) novelty detection by novel object recognition test (2); and 4) interleukin (IL)-6 concentration in the serum and cerebral cortex as a surrogate marker for inflammation.

Q2

-Page 8, Line137-139 and Page 18, Line314-318: It is not clear to me how the parameters assessed in the open field test are related to motivation.

Ans. 2) 

We removed the word, motivation, from the indicated sentence adopting the point raised by the reviewer.

Q3

-I suggest dividing the open field test data into 5 min bins as the anxiety/activity measures are more reliable at the beginning of this test.

Ans. 3) 

We made a new table by analyzing the first 5-min data of the open field test following the reviewer’s comments. The new table (for the first 5-min) replaced the old one (for entire 10-min) in the text; the “10-min” table is presented as the S1 table. Figure panels for 6B and C also replaced ones for the first 5-min. Relevant correlations were essentially conserved for the first 5-min window analysis, with a couple of additional statistical significances. We carried out the third set of analysis for the second 5-min window data to find above mentioned significant correlations were limited to the first 5-min.

Q4

-Regarding the correlation between the behavioural parameters and oscillation power. It would be interesting to do correlation separately within control and LPS group as the direction of the correlation can change in control vs LPS.

Ans. 4) 

We did a set of new analysis after separating control- and LPS-mice following the suggestion by the reviewer. Indeed, we noted a statistical significance in a data subset, i.e. theta band oscillatory power v.s. spent time in the center area, specifically for the control-group, which was not evident in the combined data set. A further look reveled that one or two data points in the control-group had far stronger statistical influence over the remaining ones, raising the possibility of a false-positive case. We concluded that it would be safer not to separate control- and LPS-groups for the current data set, considering the limited number of animals used in this study. We attached a scatter diagram for this answer. See answers to reviewers at the end of the revised pdf file (page 86). 

Q5

-For the correlation, the time spent in the centre is not reported. I think this is a better readout then the entries. Also, the correlation should be performed with the data corresponding to the first 5 min of the open field as mentioned above.

Ans. 5) 

The time spent in the center has been reported in the center part of the Table 1. We admit the column titles lacked clarity. We modified the column titles “total entries” and “total time” to “total entries to the center area” and “total time spent in the center area”.

Q6

There are still some minor grammatical issues that can be corrected during the proofreading stage.

Ans. 6) 

We are sorry for the grammatical errors remaining in the text. We carried out an extra round of checking to get rid of the errors as much as possible.

---

## [Editor Report · Decision Letter 2]

5 Apr 2024

Effects of systemic inflammation on the network oscillation in the anterior cingulate cortex and cognitive behavior

PONE-D-23-39823R2

Dear Dr. Hojo,

We’re pleased to inform you that your manuscript has been judged scientifically suitable for publication and will be formally accepted for publication once it meets all outstanding technical requirements.

Kind regards,

Kenji Hashimoto, PhD

Section Editor

PLOS ONE